# Detection of Chromosomal Segments Introgressed from Wild Species of Carrot into Cultivars: Quantitative Trait Loci Mapping for Morphological Features in Backcross Inbred Lines

**DOI:** 10.3390/plants11030391

**Published:** 2022-01-30

**Authors:** Chenggang Ou, Tingting Sun, Xing Liu, Chengjiang Li, Min Li, Xuewei Wang, Huaifu Ren, Zhiwei Zhao, Feiyun Zhuang

**Affiliations:** 1Key Laboratory of Horticultural Crop Biology and Germplasm Innovation, Ministry of Agriculture, Institute of Vegetables and Flowers, Chinese Academy of Agricultural Science, Beijing 100081, China; ouchenggang@caas.cn (C.O.); yixuansmile@163.com (T.S.); liuxing02@caas.cn (X.L.); li1504850091@163.com (M.L.); 18730285257@163.com (X.W.); zhaozhiwei@caas.cn (Z.Z.); 2Suzhou Academy of Agricultural Science, Suzhou 234000, China; sznks@163.com (C.L.); renhfsz@163.com (H.R.)

**Keywords:** carrot, morphological traits, introgression, backcross inbred lines, quantitative trait locus (QTL)

## Abstract

Cultivated carrot is thought to have been domesticated from a wild species, and various phenotypes developed through human domestication and selection over the past several centuries. Little is known about the genomic contribution of wild species to the phenotypes of present-day cultivars, although several studies have focused on identifying genetic loci that contribute to the morphology of storage roots. A backcross inbred line (BIL) population derived from a cross between the wild species *Daucus carota* ssp. *carota* “Songzi” and the orange cultivar “Amsterdam forcing” was developed. The morphological features in the BIL population became more diverse after several generations of selfing BC_2_F_1_ plants. Only few lines retained features of wild parent. Genomic resequencing of the two parental lines and the BILs resulted in 3,223,651 single nucleotide polymorphisms (SNPs), and 13,445 bin markers were generated using a sliding window approach. We constructed a genetic map with 2027 bins containing 154,776 SNPs; the total genetic distance was 1436.43 cM and the average interval between the bins was 0.71 cm. Five stable QTLs related to root length, root shoulder width, dry material content of root, and ratio of root shoulder width to root middle width were consistently detected on chromosome 2 in both years and explained 23.4–66.9% of the phenotypic variance. The effects of introgressed genomic segments from the wild species on the storage root are reported and will enable the identification of functional genes that control root morphological traits in carrot.

## 1. Introduction

Wild carrot (*Daucus carota* L. ssp. *carota*) is a widespread weedy species that is distributed in Europe, Southwest and Central Asia, North Africa, North and South America [1,2], and found from Afghanistan to the Mediterranean area [3]. Approximately 5600 carrot and *Daucus* species accessions have been collected and preserved worldwide [1]. The purple and yellow carrot cultivars (*D. carota* L. ssp. *sativus*) can be regarded as early ancestral types from approximately 1100 years ago [4,5], the red or purple-red cultivars are recorded from about 440 years ago [6], and orange cultivars are not reliably reported until the sixteenth century in Europe [7]. Over the last few centuries, the carrot has become one of the most important and nutritional Apiaceae crops. Various cultivars with orange roots have been developed to meet consumer demands; examples are “Nantes”, “Kuroda”, “Imperator”, and “Amsterdam forcing” [1]. Recent molecular evidence suggested that wild species have made significant contributions to carrot evolution and domestication [4,5,6,8,9,10]. Gene flow between cultivated and wild carrot has resulted in domesticated carrot sharing many of the same alleles as wild carrot [11]. An open question is to what extent the genome segments originating from wild carrot species affect phenotypes in the cultivars. At present, little is known about the roles that wild species have played in the domestication of carrot.

Marker-assisted selection (MAS) has been proven to be an effective tool for crop breeding and genetic studies of target traits in plants [12,13], but the efficiency is highly dependent on the precise quantitative trait loci (QTLs) or markers [14,15]. A high-resolution genetic map can greatly improve the efficiency of QTL detection, discovery of potential functional genes, and assembly of genome sequence. The density of genetic maps can be improved by developing polymorphic molecular markers that are distributed throughout the genome using high-throughput sequencing technology [16]. Sequencing with low coverage is one of the most economical and effective methods for rapid development of many genome-wide single nucleotide polymorphism (SNP) markers [17,18]. A sliding window strategy, which combines multiple SNPs with the same genotype into a bin marker with polymorphism, has been successfully applied to the construction of genetic maps and to QTL analysis [15,16,17,19]. The carrot genome sequence has been published [5]. Consequently, several high-throughput genotyping-based platforms were used to develop sequence-based SNP markers to construct high-density linkage maps and discover key genes or QTLs related to storage root traits such as carotenoid accumulation, anthocyanin content, root development, and biomass [5,20,21,22,23,24].

Growth of the carrot storage root is controlled by a sophisticated regulatory network which consists of multiple endogenous and external factors. The heritability of root width and length is controlled by additive variance and the interaction between additive variance and environmental variance [25]. The root length increases most at low temperatures, and the lower part of the root thickens slowly at moderate temperatures [26]. Increasing plant density is accompanied by a decrease in root length and the development of a cylindrical shape in young roots [27]. Furthermore, root growth and development can be regulated by endogenous hormones [28]. For example, auxins can alter root development in a tissue-specific and stage-dependent manner [29]. Gibberellin not only inhibited root enlargement and stimulated shoot growth but was also found to enhance lignification in the secondary xylem [30,31]. Recently, five QTLs were detected that relate to root thickening, and a candidate gene, *DcAHLc1*, was proposed to regulate root development [22]. Numerous QTLs have been identified in the carrot genome that are involved in regulating shoot height, biomass, and petiole number; for example, root length and biomass QTLs were identified on chromosomes 1, 2, and 7 [23]. A 180 kb region on chromosome 1 was found to be related to root shoulder diameter; this region explained 10% of the phenotypic variance and was thought to control secondary growth of the root [32].

Backcross inbred lines (BILs) can assess the phenotypes of plants with genetically identical genes or loci and improve the accuracy of phenotyping without increasing the effort of genotyping. More importantly, BILs can be used to analyze the interactions between genetic and environmental factors [33]. In this study, a BIL population derived from crossing *D. carota* ssp. *carota* “Songzi” (Ws) with an orange cultivar *D. carota* ssp. *sativus* “Amsterdam forcing” (Af) was used to study the genetics of carrot morphological traits. The root features of the BC_1_F_1_ and BIL plants will allow us to evaluate the effect of wild species on the evolution and domestication of carrot. The morphology of the BIL population plants was investigated and compared for two years, and stable QTLs related to them were detected with a high density bin map. This study will provide new insights into our understanding of the influence of genetic and environmental interactions on morphological traits and contribute to the identification of putative functional genes. Furthermore, it will be possible to illustrate the genetic mechanisms of root shape and size of carrot in future.

## 2. Materials and Methods

### 2.1. Plant Materials and Morphological Evaluation

*D. carota* ssp. *carota* “Songzi” (Ws) collected from Hubei province does show a biennial growth habit in nature [2], but we found that it is sensitive to bolting and flowering under long-day conditions even without low temperatures, and it can complete its reproductive cycle from seed to seed in 4–6 months as an annual [34,35]. *D. carota* ssp. *sativus* “Amsterdam forcing” (Af) is an orange-rooted European cultivar (Western type) [1]. An F_1_ hybrid was produced by crossing the wild species Ws with the cultivar Af. The BC_1_F_1_ generations were generated by backcrossing the F_1_ to the Af parent using artificial emasculation, and only 65 of 80 offspring were left to generate BC_2_F_1_. Four or five individuals from each BC_2_F_1_ were randomly screened as the candidates (302 BC_2_F_1_ plants) and were then self-pollinated for six generations through single-seed descent (SSD) to develop a set of 223 BC_2_S_6_ plants that together comprised the backcross inbred line (BIL) population. Seeds from 110 BILs were sown in the field with a spacing of 5 × 15 cm at the Changping station of the Chinese Academy of Agricultural Science in late July, and the plant morphology was examined in mid-November in 2014 and 2015.

Root color, root length (RL), and root shoulder diameter (the maximum diameter of the root shoulder, RSD) of the BC_1_F_1_ plants were investigated. Nine measured traits and seven indexes were used to describe the morphology of the BIL population at the vegetative stage [3]. The maximum leaf length (MLL) and leaf weight (LW) were measured to discover the genetic loci that contribute to foliage biomass. RL, RSD, root middle diameter (RMD), root tip diameter (diameter of root tip above the 1 cm position, RTD), root weight (RW), plant weight (weight of the leaves and root, PW), and dry matter content of the root (DMC) were measured to analyze root biomass and the inheritance of these characters. Seven indexes, LW/RW (iLR), RSD/RMD (iSM), RSD/RTD (iST), RMD/RTD (iMT), RSD/RL (iSL), RMD/RL (iML), and RTD/RL (iTL), were calculated to analyze root shape and biomass distribution. All traits were evaluated using the methods described by IPGRI [3]. Five plants of each BIL were randomly sampled to evaluate the forementioned traits in three biological replicates. This investigation was repeated in 2014 and 2015. Analysis of the variance (ANOVA) and the Pearson correlation coefficients among the traits were determined with SPSS software (version 10.0; SPSS, Chicago, IL, USA).

### 2.2. Genome Resequencing of the BIL Population

Leaves were sampled from five plants of each BIL and the two parents, separately, were immediately frozen in liquid nitrogen and stored at −80 °C. Total genomic DNA was extracted using the cetyltrimethylammonium bromide (CTAB) method with modification as described in Briard [36]. High-throughput genome sequencing of each sample was performed on an Illumina HiSeq 2500 sequencing platform at Biomarker Technologies Co. Ltd. (Beijing, China). The clean reads were filtered by removing adapters, reads containing >10% Ns (unknown bases), and reads in which more than 50% of the bases had a *Q* score less than 10. The 100 bp paired-end reads were aligned to the carrot genome using BWA software (http://bio-bwa.sourceforge.net/ (accessed on 1 January 2022)) [5,37]. The sequencing data were available under National Center for Biotechnology Information (NCBI) project PRJNA741874, accession numbers SRR14965956 (Af) and SRR14965957 (Ws). After filtering, the properly and uniquely mapped reads were used to call single nucleotide polymorphisms (SNPs) using the Genome Analysis Toolkit (GATK) with the default parameters [38].

### 2.3. SNP Identification and Mapping in the BIL Population

The SNPs were further filtered using the following criteria: (1) SNPs were homozygous with two parental genotypes, and there were at least 60 BILs with SNPs at a single site; (2) the SNP depth of the two parents was more than fourfold. A modified sliding window approach, as described by Huang et al. [17], was used to identify the genotype and recombination bin. The genotype of each window was called with a window size of 50 bp and a step size of 1 bp and determined based on the ratio of SNPs with genotype of two of the parents: when more than 2/3 of the SNPs had one parental genotype, the window was considered homozygous; otherwise, the window was considered as heterozygous. Adjacent 50 kb intervals without recombination breakpoints in all 110 BILs were combined into one recombination bin; bins were excluded using the chi-square test with a *p* value less than 0.005 [17]. The genetic map was constructed with JoinMap version 4.0 (https://www.kyazma.nl/index.php/JoinMap/ (accessed on 1 January 2022)) with default parameters and the genetic distances between bins were calculated using the Kosambi mapping function [39]. Linkage groups were oriented and numbered by aligning resequencing data to the carrot genome and corresponded to nine chromosomes [5].

### 2.4. QTL Analysis

The morphological traits and seven indexes were detected with the composite interval mapping (CIM) approach using Windows QTL Cartographer (WinQTLCart) version 2.5 [40]. The parameters were followed for Model 6: forward–backward stepwise regression as the background control, scanning window size 10.0 cm, walk speed 1.0 cm and the logarithm of the odds (LOD) threshold value determined with 1000 permutations at a significance level of 0.05, and a 95% confidence interval in the QTL region [40].

## 3. Results

### 3.1. Morphology of the F_1_ Hybrid and the Backcross Progeny Plants

The root of the wild species (Ws) accession SRR14965957 (*D. carota* ssp. *carota* “Songzi”) is white, conical in shape, inedible, and lignified with deep lateral root scars and tiny lateral roots, and the plants complete their reproductive cycle as annuals [35]. Cultivar Af has a biennial growth habit, and its root is orange and cylindrical in shape with a small purple/green shoulder area (Figure 1A). Some morphological traits of the F_1_ hybrid such as leaf number, white root, conical shape, deep lateral root scars, rough skin, and annual growth habit were clearly inherited from the Ws parent [34], but other traits showed obvious heterosis, such as MLL and RL (Figure 1A,D). Most of the BC_1_F_1_ plants showed inheritance of traits from the recurrent Af parent, such as cylindrical roots with smooth skin, but a few of the plants retained traits from Ws such as premature bolting or many lateral roots (Figure 1B). It was noteworthy that the BC_1_F_1_ plants showed diversity in terms of root color that ranged from white (10 plants), to pale yellow (21), yellow (25), pale orange (14), and orange (10). RL in the BC_1_F_1_ plants was 7.6–22.3 cm, and RSD was 1.9–5.2 cm (Figure 1D). After 65 of the BC_1_F_1_ plants were backcrossed with the Af parent again, 302 BC_2_F_1_ plants were randomly screened to establish the BIL population through single-seed descent (SSD). In the BC_2_S_2_ generation, the color, shape, and size of the storage roots became more diverse, and root color, in particular, gradually changed from yellow to orange, but there were no plants with purple roots (Figure 1C).

After self-pollination for six generations of SSD, 223 BC_2_S_6_ lines (the BILs) were obtained and they showed a high level of morphological variation in traits such as root color, root skin, RL, RSD, and green shoulder (Figure 1E). Most lines possessed the biennial habit of the Af parent, but four lines (E3609, E4501, E4503, and E5201) maintained the annual habit of the Ws parent. Representative lines from the 110 BILs could be divided into four groups based on RL, root skin, root shape, and green shoulder. In group a, the lines had roots with a conical shape, deep lateral root scars, and rough skin that were inherited from Ws. In group b, the lines had much greener shoulders that were different from either of the parents. In group c, the roots were short and conical in shape with sharp tip roots, while in group d, the roots had smooth skins, were cylindrical in shape, and had rounded root tips inherited from Af. Moreover, some intermediate root colors were not found as described in a previous report [6]. For example, lines E0909, E0501, and E4608 had pale orange roots with a low β-carotene content (data not shown; Figure 1E).

### 3.2. Variation in Morphological Traits in the BIL Population

Nine measured traits and seven indexes were used to describe the morphological variation present in the BIL population in 2014 and 2015 (Figure 2). The average values for MLL (54.3/54.7 cm), RL (17.2/17.4 cm), DMC (12.4/11.5%), iSM (1.21/1.21), and iLR (0.37/0.36) were similar for the two years. The average values for LW, RSD, RMD, RTD, RW, PW, iML, and iTL in 2014 were significantly lower than in 2015, while the values for iST and iMT were higher in 2014 than in 2015. The range of variation for LW, RSD, RMD, RTD, RW, PW, iST, iMT, and iTL were different in the two years.

The correlation coefficients of the nine measured traits in the BIL population are shown for 2014 and 2015 (Table 1). In 2014 and 2015, LW was positively correlated with MLL (0.62 ***/0.46 ***) and RL (0.26 **/0.57 ***), but MLL and RL were not correlated; RL was positively correlated with RSD (0.22 */0.51 ***), RW (0.46 ***/0.70 ***), and PW (0.48 ***/0.72 ***); PW showed a positive relationship with RSD (0.83 ***/0.88 ***), RMD (0.80 ***/0.83 ***), RTD (0.35 ***/0.63 ***), and RW (0.96 ***/0.98 ***); DMC was negatively correlated with RMD (−0.44 ***/−0.30 **) and RTD (−0.40 ***/−0.37 ***).

### 3.3. Genotyping and Construction of the Bin Map in the BIL Population

Genomic DNA from the two parents was sequenced, yielding 5.6 Gbp of clean bases with a depth of approximately tenfold. For Ws and Af, 95.74% and 97.03% of the total reads could be aligned to the carrot reference genome and covered 88.62% and 93.3% of the genome, respectively (Appendix A). Each BIL was sequenced to approximately 1.8-fold depth. For each BIL, 85.91–97.92% of the reads were aligned to the reference genome with 44.46–72.41% coverage. After analysis using the GATK software package, a total of 3,223,651 SNPs were identified, and 333,280 homozygous SNPs were polymorphic between Ws and Af. The ratios of heterozygous SNPs of Ws and Af were 50.81% and 72.17%, respectively. The average ratio of heterozygous SNPs of the BIL population was 18.88% and were in the range 7.52–34.31% (Appendix A). After filtering, a total of 259,246 recombination breakpoints were detected with an average of 33.2 breakpoints per line. In total, 13,445 bins were generated and used to construct the genetic linkage map. The physical lengths of the bins were in the range 20.04 kb–4.21 Mb with an average length of 130.88 kb.

A high-density genetic map was constructed by mapping 2027 bins onto the nine carrot chromosomes with an average of 219.6 bins per chromosome (Table 2; Appendix A). The map consisted of 154,776 SNPs with an average of 76.4 SNPs in each bin. The total recombinational length of the linkage map was 1436.43 cm, with an average interval length of 0.71 cm, which corresponded to a physical distance of 0.33 Mb/cm over the whole carrot genome [5]. The largest number of bins was on chromosome 4:281 bins containing 22,429 SNPs, with an average distance of 0.60 cm between adjacent bins covering a genetic length of 167.26 cm. Chromosome 8 had the smallest number of bins (141 containing 5564 SNPs), and it had the largest average genetic distance of 1.18 cm between bins for a genetic length of 166.27 cm. Large gaps were present on each chromosome, and the largest gap was 16.88 cm on chromosome 2.

We found that small segments of the Ws genome were randomly distributed on each chromosome in most cases (Figure 3) and the percentage of homozygous bins from Ws ranged from 3.5% (chromosome 2) to 9.7% (chromosome 1) with an average of 5.8% (Appendix A). For each BIL, 76.4–99.4% of the homozygous bins were from the Af parent, while those from the Ws parent comprised 0.6–23.6% (Figure 3, Appendix A). For example, 0.6% of the genome of E2802, a line with a pale orange root with a cylindrical shape and round tip (Figure 1E(d)), came from Ws and 99.4% came from Af (Figure 3a); the E5201 line had a small white root and an annual growth habit (Figure 1E(a)), with 23.6% of the genome coming from Ws and 76.4% from Af (Figure 3b). However, notably, many genomic segments originating from Ws were clustered at the terminus of chromosome 1 in most of the BILs, and some large segments were also maintained in some BILs. For example, line E1202 has a large Ws genomic segment on chromosome 9 but has a pale orange root with smooth skin (Figure 3c); 52.21% of chromosome 2 in line E5302 came from the Ws genome (Figure 3d), and the plants have small white storage roots and rough skin (Figure 1E(a)).

### 3.4. QTL Analysis of BIL Morphological Traits

QTLs for the nine morphological traits and seven indexes were detected in the bin map with WinQTLCart using the CIM method [40]. There were 23 and 25 QTLs identified in 2014 and 2015, respectively, and only five QTLs were found to be stable in both years (Table 3 and Figure 4). Five, 16, 4, 2, 3, 2, 3, 2, and 1 QTLs were located on chromosomes 1 to 9, respectively, and explained 10.1–66.9% of the phenotypic variation (Table 3). One QTL was detected for MLL (maximum leaf length), *MLL2-1*, in 2014 and two QTLs, *MLL1-1* and *MLL5-1*, were detected in 2015, which explained 25.7, 13.2, and 13.9% of the phenotypic variation, respectively. Both *MLL2-1* and *MLL1-1* had positive additive effects. Three QTLs, *LW1-1*, *LW1-2*, and *LW9-1*, were detected for LW (leaf weight) in 2014 and two QTLs, *LW2-1* and *LW2-2*, were detected in 2015; these five QTLs explained 12.7, 17.3, 15.2, 13.1, and 16.6% of the phenotypic variation, respectively. *LW1-2* had a positive additive effect, while the other four QTLs had negative additive effects. For RL (root length), *RL2-1* was stable in that it was identified in both years and explained 33.7% and 30.5% of the phenotypic variation in 2014 and 2015, respectively, and *RL2-2* explained that 40.4% of the phenotypic variation was identified in 2015. Both *RL2-1* and *RL2-2* had positive additive effects. For RSD (root shoulder diameter), *RSD2-1* was detected in 2014 and explained 39.1% of the phenotypic variation, while another QTL, *RSD2-2*, was stably detected in both years and explained 39.1% and 24.6% of the phenotypic variation, respectively. For RMD (root middle diameter), the QTLs *RMD3-1* and *RMD2-1* were identified in 2014 and 2015, respectively, and explained 38.9% and 29.5% of the phenotypic variation. For RTD (root tip diameter), only one QTL, *RTD2-1*, was identified in 2014, and explained 12.6% of the phenotypic variation. For RW (root weight), two QTLs were identified in each year that were located on four chromosomes. *RW1-1* and *RW5-1* had positive additive effects and explained 10.5% and 13.2% of the phenotypic variation, respectively, while *RW7-1* and *RW8-1* had negative additive effects and explained 11.4% and 15.9% of the phenotypic variation in 2014 and 2015, respectively. Three QTLs, *PW1-1*, *PW2-1*, and *PW4-1*, were identified for PW (plant weight) in 2014 and explained 13.2, 18.9, and 15.3% of the phenotypic variation, respectively, but only one QTL, *PW8-1*, was identified in 2015, and it explained 14.1% of the phenotypic variation. *PW1-1* and *PW2-1* had positive additive effects, but *PW4-1* and *PW8-1* had negative additive effects. Only one QTL, *DMC2-1*, was identified for DMC (dry matter content of the root) in both 2014 and 2015 and explained 23.4% and 37.2% of the phenotypic variation, respectively. In addition, *DMC2-2* and *DMC5-1* were identified in 2015 and explained 37.2% and 32.4% of the phenotypic variation, respectively.

Three QTLs were identified for iSM (index of RSD/RMD) in 2014. *iSM2-1*, *iSM2-2*, and *iSM3-1* explained 66.9, 66.9, and 24.9% of the phenotypic variation, respectively. *iSM2-1* and *iSM2-2* had positive additive effects, but *iSM3-1* had a negative additive effect. Notably, *iSM2-1* and *iSM2-2* were stably detected in 2015 and explained 55.7% and 55.7% of the phenotypic variation, respectively. Two QTLs, *iST7-1* and *iST5-1*, were identified for iST, the index RSD/RTD, in 2014 and 2015, respectively, and explained 30.9% and 27.4% of the phenotypic variation, respectively. One QTL, *iMT7-1*, was identified for iMT, the index RMD/RTD, in 2014 and explained 31.9% of the phenotypic variation and had negative additive effects. Two QTLs, *iMT2-1* and *iMT2-2*, were identified in 2015 and explained 36.3% and 36.3%, respectively of the phenotypic variation. Two QTLs for iSL, the index RSD/RL, were identified, *iSL3-1* and *iSL2-1*, that explained 30.1% and 31.2% of the phenotypic variation in 2014 and 2015, respectively. Only one QTL, *iML3-1*, was identified for iML, the index RMD/RL, in 2014, and explained 16.5% of the phenotypic variation. Three QTLs, *iTL2-1*, *iTL2-2*, and *iTL2-3*, were identified for iTL, the index RTD/RL, in 2015 and explained 27.6, 22.8, and 12.3% of the phenotypic variation, respectively, but none of them were identified in 2014. A single QTL, *iLR3-1*, was identified for iLR, the index LW/RW, in 2014 that explained 15.7% of the phenotypic variation, and two QTLs, *iLR6-1* and *iLR6-2*, were identified in 2015 that explained 16.0% and 18.7% of the phenotypic variation, respectively. Five QTLs (*RL2-1*, *RSD2-2*, *DMC2-1*, *iSM2-1*, and *iSM2-2*) were consistently detected in both years and were thus considered to be stable QTLs. Mostly, Af contributed positive additive effects for each QTL, while Ws contributed positive additive effects for QTLs related to MLL, LW, DMC, and iLR.

It was notable that *RL2-2*, *RSD2-1*, *DMC2-2*, *iSM2-1*, and *iMT2-1* were clustered together in a 35.2 cm region of chromosome 2 and *MLL2-1*, *RL2-1*, *RSD2-2*, *RMD2-1*, *PW2-1*, *DMC2-1*, *iSM2-2*, *iMT2-2*, and *iSL2-1* were clustered together in a 44.9 cm region of chromosome 2 (Table 3; Figure 4).

## 4. Discussion

Cultivated carrot (*Daucus carota* L. ssp. *sativus*) is the only subspecies with an edible root in the genus *Daucus* and has been suggested to be domesticated from the eastern wild species *D. carota* L. ssp. *carota* [4,41]. A significant signature of selection upon domestication showed that development of the large storage root was a transition from the wild species to the cultivated carrot [22]. After domestication and selection, the cultivated carrots differ in many morphological characters compared to the wild species, including increased size and variation in root shape, loss of lateral root branching, increased carotenoid and anthocyanin contents, and a biennial growth habit [42]. However, little is known about the effects of genome segments from the wild species during carrot domestication, especially for the mechanism of expansion of the storage root. MLL and RL in the F_1_ hybrids showed obvious heterosis, while root color, conical root shape, deep lateral root scars, and the annual growth habit of the F_1_ were inherited from the Ws parent (Figure 1A,D). It was interesting that the BC_1_F_1_ plants showed variation in root color, shape, and size, and only a few plants retained the annual growth habit or lateral roots from Ws (Figure 1B). The annual habit and/or presence of lateral roots may give these plants a reproductive advantage, but it was difficult for them to complete their reproductive cycles from autumn to winter. Thus, only 65 root stocks were left in the next year. The root color, shape, and size in the BC_2_S_2_ population become more diverse, although none were purple in color (Figure 1C), a trait that can be stably transmitted to the BIL lines (Figure 1E). Some BIL root colors have not even been reported in carrot landraces, such as pale orange (Figure 1E). Most BIL lines appeared to have the biennial growth habit inherited from the Af parent, and only four lines retained the annual habit of Ws, which indicates that an annual growth habit may be a disadvantageous character for wild carrot species to survive, especially in the north of China. These results could explain why *D. carota* ssp. *carota* can complete the reproductive cycle as an annual but still retains the biennial growth habit and has thus spread widely in southern China [2]. Moreover, Chinese carrot landraces as the eastern type were more sensitive to premature bolting in early spring than the western varieties with orange roots [4,6], which suggests that these landraces have retained the annual growth habit from their progenitor to some degree. The annual growth habit is a disadvantageous character for commercial cultivars during the winter–spring period or spring cultivation; however, it is advantageous in that it enables these landraces to survive. This explains why nonadapted phenotypes may have had limited gene flow from the wild species, but a limited domestication bottleneck between wild and cultivated carrots [4,5]. During long-term domestication, the morphological traits were continuously selected in response to consumer needs by breeders, which resulted in local reductions in genetic variation of domestication genes in the cultivated populations compared to their wild counterparts [4,22].

Carrot root shape and size were the major commercial traits that distinguished the various types in the development of the orange carrot, which progressed due to selection since the 1600s [43]. In this study, nine morphological traits and seven indexes were used to analyze variation in the BIL population over two years. Average values for MLL, RL, DMC, iSM, and iLR appeared to be stable while the others showed significant differences between the two years (Figure 2). In particular, larger root diameters in 2015 (including RSD, RMD, and RTD) indicated that the roots were much thicker, and thus resulted in higher individual root weights based on the higher correlation coefficients between RW and RSD and between RW and RMD (Table 1).

QTL mapping provides more information to understand the genetic mechanisms for morphological traits. In this study, a total of 43 QTLs were detected using a high-density bin map for 16 morphological traits in carrot, while five out of them (*RL2-1*, *RSD2-2*, *DMC2-1*, *iSM2-1*, and *iSM2-2*) were stable in the two years. Most of the QTLs were only detected in one year. The same situation has been observed in other crops [19,44,45]. It indicates that the inheritance of morphological traits is affected by the interaction between genetic and environmental factors and is controlled by various genes under different environmental conditions. Thus, QTLs that were consistently identified in different environments can provide confirmation that these genetic loci can be reliably used to discover candidate genes in carrot. The four stable QTLs for root length and width (*RL2-1*, *RSD2-2*, *iSM2-1*, and *iSM2-2*) were located on the same chromosome as previously reported [22,23], and we can infer that the heredity of storage root shape and size in carrot is possibly controlled by candidate genes on chromosome 2. However, Macko-Podgórni et al. identified a 180 kb long region that was responsible for RSD on chromosome 1 [32]. These results indicated that the development of the storage root and expansion may be regulated by multiple genes. In addition, two clusters of QTLs were identified on chromosome 2. One was located in a 34.2–35.6 cm region compressing five QTLs related to RL, RSD, DMC, iSM, and iMT. The other was located in a 44.4–45.0 cm region compressing nine QTLs related to MLL, RL, RSD, RMD, PW, DMC, iSM, iMT, and iSL. These traits were partially correlated with each other at significant levels (Table 1), and represented different aspects of root shape, size, and biomass in carrot. For example, RL, RSD, RMD, iSM, iMT, and iSTL were the different components of root shape and size, while PW and DMC represented root biomass. Colocalization of QTLs for the correlated traits suggests the existence of pleiotropic linkage and similar regulating mechanisms. The phenomenon of genetic separation could effectively improve the utilization of genes and reduce gene loss resulting from genetic recombination [44,46,47].

Cultivated carrot phenotypes are distinguished based on their root shapes and sizes. Several root morphological traits of interest to breeders, such as root length, root width, and biomass, were previously evaluated by means of labor-intensive measurement or image analysis [23]. In this study, a BIL population was used to examine the influence of genome introgression of the wild species on carrot storage root development. The root colors, shapes, and sizes in the BIL population became more diverse. Few lines retained an annual growth habit or lateral roots as wild parent, while most lines displayed an expanded storage root and biennial growth habit as cultivar parent. Five stable QTLs that contributed to four root traits were identified consistently in two years. The results presented here will provide more useful clues for the discovery of functional genes in future research and the development of molecular markers for selective breeding in carrot.

## Figures and Tables

**Figure 1 plants-11-00391-f001:**
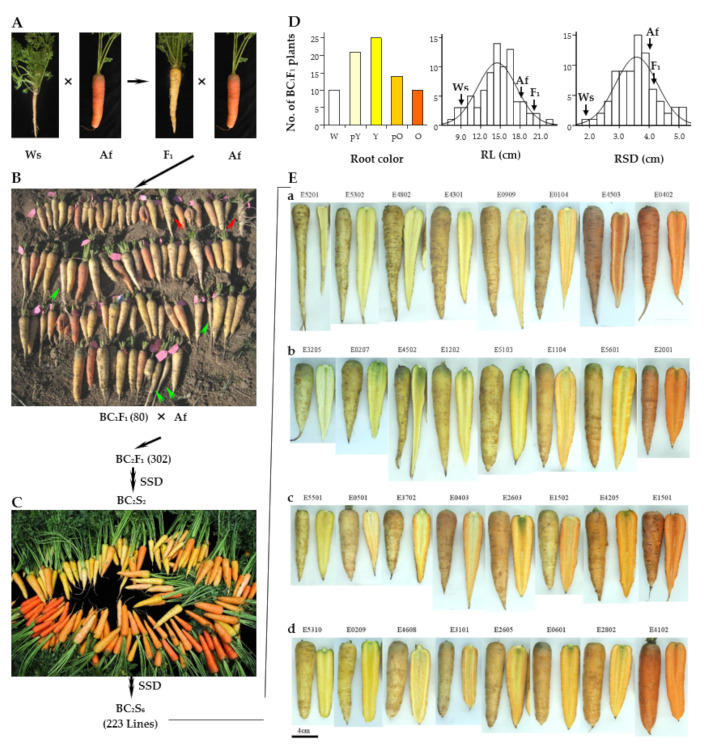
Construction of a population of backcross inbred lines. (**A**) The morphology of the parents, Ws and Af, and their F_1_. Ws is the wild species with white and conical shape root, deep lateral root scars, and tiny lateral roots. Af is the orange cultivar with cylindrical shape and smooth skin root. The root of F_1_ is white, has a conical shape, and rough skin with deep lateral root scars. (**B**) The morphology of the BC_1_F_1_ population with 80 plants. About 65 BC_1_F_1_ plants were left to backcross with the recurrent Af parent. The red arrows show the root with many lateral roots and the green arrows show the premature bolting plants. (**C**) The morphology of the BC_2_S_2_ population. SSD, single seed descent. (**D**) Root color, RL, and RSD distribution in the BC_1_F_1_ population. W: white; pY: pale yellow; Y: yellow; pO: pale orange; O: orange. Black arrows represent the means of Ws, Af, and F_1_. (**E**) The morphology of partial BC_2_S_6_ lines (BILs). Representative lines from 110 BILs could be divided into four groups according to RL, root skin, shape, and green shoulder. (**a**) The lines were long; the roots were conical in shape with deep lateral scars and rough skin. (**b**) The lines had a large green shoulder. (**c**) The lines were short; the roots were conical in shape and had sharp tips. (**d**) The lines were short; the roots were cylindrical in with smooth skin. Bar is 4 cm.

**Figure 2 plants-11-00391-f002:**
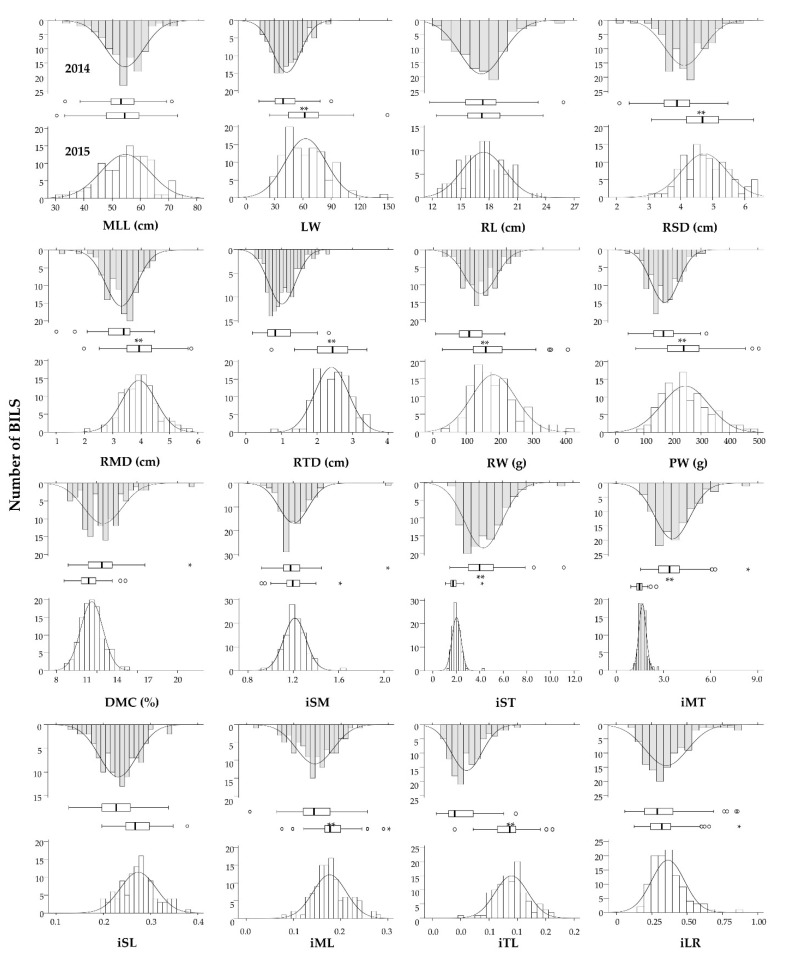
Comparison of the morphological traits in the BIL population. Downward-facing gray histogram shows the distribution of each trait in 2014, and upward-facing white histogram shows the distribution in 2015. The middle boxplots and average values of each trait were analyzed in 2014 and 2015 with ANOVA by SPSS software. * and ○ mean outliers in the boxplot; ** beside white boxes mean the average values of each trait at a significance level of 0.01.

**Figure 3 plants-11-00391-f003:**
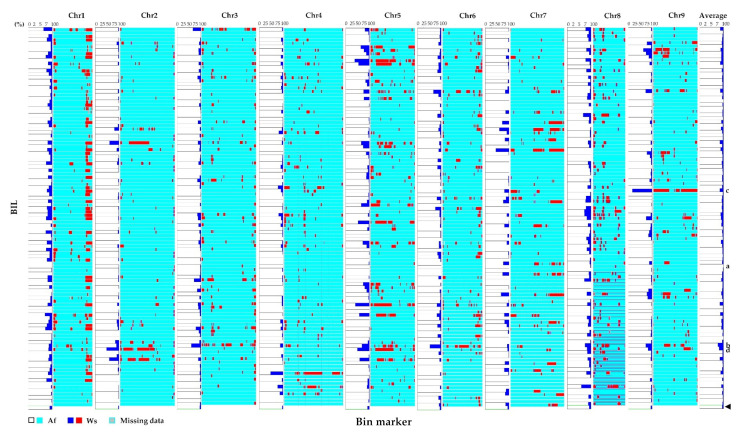
Recombination bin map of the BIL population. Each line represents bin markers in the order of the linkage map. White and blue columns represent the ratio of homozygous bins from the Af and Ws genomes on the chromosome, respectively. Light blue and red columns represent the genetic regions of homozygous bins from the Af and Ws genomes on the chromosome, respectively. (**a**) The E2802 line had a pale orange root, cylindrical shape, and round tip, with 0.6% and 99.4% of the genome coming from Ws and Af, respectively. (**b**) The E5201 line had a small white root and annual growth habit, with 23.6% and 76.4% of the genome from Ws and of Af, respectively. (**c**) The E1202 line had a large genome fragment from Ws on chromosome 9 with pale orange root and smooth skin. (**d**) The E5302 line had 52.21% of the genome coming from Ws on chromosome 2 and presented small white roots and rough skin. The black arrowhead indicating green boxes shows average relative ratios of the Af (white) and Ws (blue) genome on each chromosome.

**Figure 4 plants-11-00391-f004:**
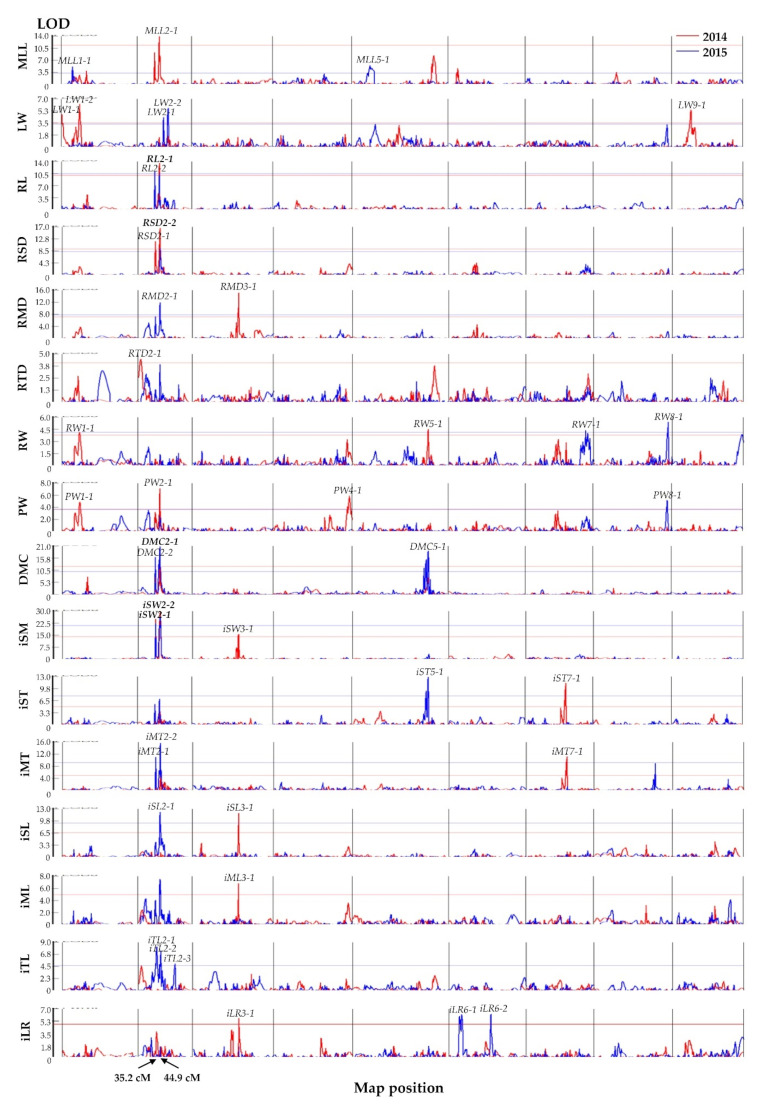
The QTLs of the morphological traits of the BIL population in 2014 and 2015. The curves indicate the genetic position (*x*-axis) of bin markers against LOD score (*y*-axis) of QTLs detected on nine chromosomes. The red and blue lines represent the LOD threshold in 2014 and 2015, respectively. Arrows indicate the position of clusters on chromosome 2. Cluster 1 was mapped at 35.2 cm including *RL2-2*, *RSD2-1*, *DMC2-2*, *iSM2-1*, and *iMT2-1*. Cluster 2 was mapped at 44.9 cm including *MLL2-1*, *RL2-1*, *RSD2-2*, *RMD2-1*, *PW2-1*, *DMC2-1*, *iSM2-2*, *iMT2-2*, and *iSL2-1*.

**Table 1 plants-11-00391-t001:** Correlation coefficients of morphological traits in the BIL population.

Traits	MLL	LW	RL	RSD	RMD	RTD	RW	PW	DMC
MLL		0.46 ***	0.18	0.06	−0.01	0.08	0.10	0.20 *	−0.21 *
LW	0.62 ***		0.57 ***	0.60 ***	0.49 ***	0.31 **	0.63 ***	0.78 ***	−0.07
RL	0.13	0.26 **		0.51 ***	0.42 ***	0.29 **	0.70 ***	0.72 ***	0.09
RSD	0.29 **	0.30 **	0.22 *		0.87 ***	0.61 ***	0.85 ***	0.88 ***	−0.17
RMD	0.25 **	0.17	0.08	0.86 ***		0.75 ***	0.85 ***	0.83 ***	−0.30 **
RTD	0.21 *	−0.04	−0.25 **	0.30 **	0.57 ***		0.41 ***	0.63 ***	−0.37 ***
RW	0.29 **	0.28 **	0.46 ***	0.88 ***	0.86 ***	0.68 ***		0.98 ***	−0.17
PW	0.44 ***	0.55 ***	0.48 ***	0.83 ***	0.80 ***	0.35 ***	0.96 ***		−0.25 **
DMC	−0.12	−0.01	0.12	−0.36 ***	−0.44 ***	−0.40 ***	−0.29 **	−0.16	

Note: The numbers below the diagonal are correlation coefficients in 2014 and the numbers above the diagonal are correlation coefficients in 2015. *, **, *** represent 0.05, 0.01, and 0.001 significance levels, respectively. Light gray, gray, and dark gray boxes show the correlation coefficients at 0.05, 0.01, and 0.001 significance levels, respectively.

**Table 2 plants-11-00391-t002:** Characteristics of the high-density genetic map.

Chr ^a^	Number of SNPs	Number of Bins	Genetic Distance (cm)	Average Distance between the Neighbor Bins (cm)	Max Gap (cm)
1	19,430	184	160.48	0.87	13.54
2	27,142	249	114.70	0.46	16.88
3	28,062	267	170.46	0.64	12.02
4	22,429	281	167.26	0.60	5.02
5	7986	224	202.93	0.91	7.96
6	12,070	194	162.37	0.84	10.60
7	17,491	250	143.02	0.58	3.93
8	5564	141	166.27	1.18	14.33
9	14,602	237	148.97	0.63	8.59
Average	17,197	225	159.61	-	10.32
Total	154,776	2027	1436.43	0.71	-

^a^ Chr indicates chromosome.

**Table 3 plants-11-00391-t003:** Identified QTLs for BIL morphological traits.

Trait	Year	QTL	Chr	LOD	Position (cm)	QTL Region (cm)	Additive Effect	Phenotypic Variance (%)
MLL	2014	*MLL2-1*	2	13.9	44.9	44.4–45.0	23.1	25.7
	2015	*MLL1-1*	1	5.2	22.8	21.8–23.6	11.8	13.2
	2015	*MLL5-1*	5	5.5	37.3	35.0–38.3	4.4	13.9
LW	2014	*LW1-1*	1	7.0	0.5	0.0–1.6	−7.2	12.7
	2014	*LW1-2*	1	6.3	37.9	36.6–40.7	11.7	17.3
	2014	*LW9-1*	9	5.6	39.4	37.5–39.9	−11.3	15.2
	2015	*LW2-1*	2	4.4	54.1	53.5–54.6	−21.9	13.1
	2015	*LW2-2*	2	5.7	64.2	63.8–64.8	−30.0	16.6
RL	2014/2015	*RL2-1*	2	13.5/10.7	44.9	44.4–45.0	8.6/8.2	33.7/30.5
	2015	*RL2-2*	2	11.1	35.2	34.2–35.6	10.7	40.4
RSD	2014	*RSD2-1*	2	12.0	35.2	34.2–35.4	2.3	39.1
	2014/2015	*RSD2-2*	2	16.6/9.7	44.9	44.4–45.0	2.3/2.2	39.1/24.6
RMD	2014	*RMD3-1*	3	15.0	96.3	95.8–96.7	1.5	38.9
	2015	*RMD2-1*	2	11.8	44.9	44.4–45.0	2.0	29.5
RTD	2014	*RTD2-1*	2	4.5	4.9	1.6–6.8	0.2	12.6
RW	2014	*RW1-1*	1	4.1	35.5	34.4–38.1	27.3	10.5
	2014	*RW5-1*	5	4.5	158.8	157.7–159.5	49.3	13.2
	2015	*RW7-1*	7	4.4	124.9	124.0–125.5	−44.5	11.4
	2015	*RW8-1*	8	5.4	155.9	155.2–156.6	−104.7	15.9
PW	2014	*PW1-1*	1	4.9	36.5	35.1–38.0	33.6	13.2
	2014	*PW2-1*	2	7.1	44.9	44.4–45.0	123.2	18.9
	2014	*PW4-1*	4	5.9	160.1	158.3–161.5	−34.2	15.3
	2015	*PW8-1*	8	5.2	154.9	154.4–156.1	−89.2	14.1
DMC	2014/2015	*DMC2-1*	2	12.3/20.9	44.9	44.4–45.0	0.1/0.1	23.4/37.2
	2015	*DMC2-2*	2	16.1	35.2	34.2–35.5	0.1	37.2
	2015	*DMC5-1*	5	18.9	159.7	158.8–159.9	−0.1	32.4
iSM	2014/2015	*iSM2-1*	2	25.1/22.3	35.2	34.2–35.4	1.1/0.6	66.9/55.7
	2014/2015	*iSM2-2*	2	29.7/27.0	44.9	44.4–45.0	1.1/0.6	66.9/55.7
	2014	*iSM3-1*	3	15.5	94.9	94.4–94.9	−0.5	24.9
iST	2014	*iST7-1*	7	11.2	84.5	84.0–84.9	−3.9	30.9
	2015	*iST5-1*	5	12.8	159.7	158.8–159.9	−1.2	27.4
iMT	2014	*iMT7-1*	7	11.2	84.5	84.0–84.9	−2.8	31.9
	2015	*iMT2-1*	2	11.0	35.2	34.2–35.4	0.9	36.3
	2015	*iMT2-2*	2	15.7	44.9	44.4–45.0	0.9	36.3
iSL	2014	*iSL3-1*	3	11.9	96.3	95.8–97	0.1	30.1
	2015	*iSL2-1*	2	12.1	44.9	44.4–45.1	0.1	31.2
iML	2014	*iML3-1*	3	6.9	96.3	95.8–97.5	0.1	16.5
iTL	2015	*iTL2-1*	2	8.2	37.9	36.7–39.0	0.0	27.6
	2015	*iTL2-2*	2	7.3	45.8	45.4–46.4	0.1	22.8
	2015	*iTL2-3*	2	5.0	76.1	74.7–79.0	−0.0	12.3
iLR	2014	*iLR3-1*	3	5.7	96.7	96.3–100.1	0.2	15.7
	2015	*iLR6-1*	6	6.3	25.1	24.1–26.9	0.1	16.0
	2015	*iLR6-2*	6	6.2	87.1	85.9–89.3	−0.2	18.7

## Data Availability

All raw reads have been deposited in the Sequence Read Archive (SRA) under project PRJNA741874, accessions SRR14965956 and SRR14965957. Information about sequence data is available at https://www.ncbi.nlm.nih.gov/bioproject/?term=PRJNA741874 (1 January 2022).

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
