# Peer review of "Detection of Chromosomal Segments Introgressed from Wild Species of Carrot into Cultivars: Quantitative Trait Loci Mapping for Morphological Features in Backcross Inbred Lines"

_plants, 2022, doi:10.3390/plants11030391_

Round 1

Reviewer 1 Report

The manuscript is written well and is of critical importance to carrot breeding and genomics researchers. However, there are some outstanding issues are noticeable such as authors have missed providing details on candidate gene analysis despite being mentioned it in the materials and methods and authors are encouraged to address them in the revised version. Additionally, there are places where different font size has been used and few typos ave been noticed in few instances. Please see below detailed comments for each section.

Reviewer 2 Report

Dear authors,

a scientifically well performed research shwoing comlex results

Comments:

lines 108-114 do not correspond with MM

Discussion

the accumulation of OTLs on chr. 2 may be more discussed

the influence of more trial years in different enviroments may discover more stable Qtls and should be discussed

Author Response

I deeply appreciated the time and effort reviewer has spent in reviewing our manuscript. These comments are helpful to improve our manuscript.

Comments #1 “lines 108-114 do not correspond with MM

Answer: Thanks for this comment. This part was tried to provide more details about the wild parent. Parent ‘Ws’ shows annual habit and sensitive to bolting and flowering after collection, while it is usually a biennial habit in nature. However, in BILs population, only few lines retained wild habit after several generations of selfing. So, this indicated annual habit may be a disadvantageous character for wild carrot species to survive. The detailed characters of Ws helped to understand the influence of genome introgression of the wild species on carrot root according to morphology.

The 1st sentence “There is only one wild carrot species, D. carota ssp. carota, that is recorded as being widely distributed in the south of China, and it appears to have a biennial growth habit [2].” (lines 108-109) was deleted in revised version.

Comments #2 “the accumulation of OTLs on chr. 2 may be more discussed

the influence of more trial years in different enviroments may discover more stable Qtls and should be discussed

Answer: Thanks for these comments. These sections were modified in revised version.

Your Sincerely

Chenggang Ou

Feiyun Zhuang

Reviewer 3 Report

The present manuscript describes on the detection of QTLs for quantitative characters through bin mapping of the regions introduced from wild carrot to cultivated one in the backcross inbred lines (BIL). The purpose of the study is very clear and the experiment is well organized. The experiment is well explained in the text, and the results and discussions are very clear. The reviewer considers that the manuscript is valuable to be accepted to the journal after minor revision.

  The reviewer gives some comments for the revision of the manuscript.

1) Title: “Role of chromosomal segments” is not understandable. Of course, it was clarified that the QTLs had some function on the expression of the quantitative characters. However, the functions of the QTLs were not analyzed here. QTLs for specific characters were detected on the chromosomes. Therefore, it might be better to change the word “Role” to “Detection”, namely, “Detection of chromosomal segments ….”

2) Abstract Line 23: The word “introgression” is not adequate to use in the present study. Introgression means introduction of genes from one species to other species via several times of natural backcrossing. In the present study, backcrossing was only twice and BILs were produced by single seed descent method.

3) Introduction: It seems that the same fact is duplicated in the first sentence (line 42-44). Therefore, it is better to describe “… and the wild Daucus species were found …” in line 44.

4) Line 70: “The carrot genome sequence has been published.” References are necessary.

5) Materials and Methods: It is necessary to describe how were the chromosome numbers 1-9 given, randomly given or based on linkage groups described in previous works?

6) Line 297: What is the meaning of ‘significance’ level of 0.01? What are the single asterisks (*), white boxes, vertical bars in the white boxes, horizontal bars and white circles?

7) Line 234: Not understandable “… in the range 20.04-4.21 Mb with an average length of 130.88kb.”

For the convenience of the authors, some detail corrections are recommended as listed below:

Abstract

Line 15: Cultivated carrot is …, and various root …. (Delete ‘The’ and ‘the’)

Line 16: … developed through human domestication …

Line 18: … contribution of …

Line 19: … the phenotypes of …

Line 22-23: The colors and shapes in the BIL population … after several generations of selfing BC2Fplants.

Line 24: … retained annual growth habit and/or lateral roots …

Line 25: displayed expanded storage roots … as the cultivated parent.

Line 29: … between the bins …

Line 35: … from the wild species …

Introduction

Line 100: … compared for two years.

Line 142: … (CTAB) method with modification as described in Briard [36].

Results

Line 225: … inbred lines. A. … of the parents, Ws and Af, and their F1.

Line 229: The red arrows show … the green arrows show …

Line 231: … distribution in the BC1F1 population.

Figure 1C: The position of the letter ‘SSD’ should be lower.

Figure 1D: No. of BC1F1 plants

Line 295: Downward facing gray histograms show the distribution of quantitative traits in 2014 and upward facing white histograms show the distribution in 2015.

Line 309: *, ** and *** represent 0.05, 0.01 and 0.001 significant levels, respectively. Dark gray, gray and light grey boxes show ….

Table 2: Average distance between the neighbor bins (cM)

Line 375: Blue and red columns represent the physical regions of the homozygous bins ….

Line 379: … arrowhead indicating green boxes shows average relative ratios of the Af (white) and Ws (blue) genomes on each chromosome.

Line 460: The QTLs of the …

Line 462: Arrows indicate …

Line 498: carrot was originated …

Line 500: … from wild species to cultivated …

Line 543: Only five out of 43 …

Line 552: …) were located on the same …

Line 563: … on their root shapes and sizes.

Author Response

Comments #1:1) Title: “Role of chromosomal segments” is not understandable. Of course, it was clarified that the QTLs had some function on the expression of the quantitative characters. However, the functions of the QTLs were not analyzed here. QTLs for specific characters were detected on the chromosomes. Therefore, it might be better to change the word “Role” to “Detection”, namely, “Detection of chromosomal segments ….”

2) Abstract Line 23: The word “introgression” is not adequate to use in the present study. Introgression means introduction of genes from one species to other species via several times of natural backcrossing. In the present study, backcrossing was only twice and BILs were produced by single seed descent method.

3) Introduction: It seems that the same fact is duplicated in the first sentence (line 42-44). Therefore, it is better to describe “… and the wild Daucus species were found …” in line 44.

4) Line 70: “The carrot genome sequence has been published.” References are necessary.

5) Materials and Methods: It is necessary to describe how were the chromosome numbers 1-9 given, randomly given or based on linkage groups described in previous works?

Answer: We deeply appreciated the time and effort reviewer has spent in reviewing our manuscript. These comments are helpful to improve our manuscript. I modified these details in the corresponding position in revised version.

The word “Role” to “Detection” in title was changed.

The sentence “after introgression of several generations” was changed to “after several generations of selfing BC2F1 plants” in abstract according to reviewer’s advice.

The sentence “the wild species is…” was deleted in introduction.

The reference of the sentence “The carrot genome sequence has been published” was labeled.

The method how to number chromosomes was supplemented in section 2.3.

Comments #2:6) Line 297: What is the meaning of ‘significance’ level of 0.01? What are the single asterisks (*), white boxes, vertical bars in the white boxes, horizontal bars and white circles?

Answer: Thanks for this comment. The boxplots between two histograms showed the distribution of each trait in different ways in figure 2. Two asterisks nearby white boxes in boxplots meant that average value of each trait was at significance level of 0.01. The single asterisk and white circles mean outlier in the boxplot. The vertical bars mean minimum, lower quartile, median, upper quartile and maximum, respectively.

Comments #3: “7) Line 234: Not understandable “… in the range 20.04-4.21 Mb with an average length of 130.88kb.

Answer: Thanks for this comment. I’m so sorry about that an unit was deleted by mistake. I change the sentence “… in the range 20.04-4.21 Mb …” to “… in the range 20.04 kb-4.21 Mb …”.

Other minor comments were modified in revised manuscript according to reviewer’s suggestions.

Your Sincerely

Chenggang Ou

Feiyun Zhuang

Round 2

Reviewer 1 Report

Authors have made significant revision to the revised manuscript and incorporated the made suggestions, I have no further comments.